# A Review of Recent Advances of Dielectric Barrier Discharge Plasma in Catalysis

**DOI:** 10.3390/nano9101428

**Published:** 2019-10-09

**Authors:** Ju Li, Cunhua Ma, Shengjie Zhu, Feng Yu, Bin Dai, Dezheng Yang

**Affiliations:** 1Key Laboratory for Green Processing of Chemical Engineering of Xinjiang Bingtuan, School of Chemistry and Chemical Engineering, Shihezi University, Shihezi 832003, China; leej222@163.com (J.L.); zsj497262724@gmail.com (S.Z.); yufeng05@mail.ipc.ac.cn (F.Y.); db_tea@shzu.edu.cn (B.D.); 2Laboratory of Plasma Physical Chemistry, School of Physics, Dalian University of Technology, Dalian 116024, China; yangdz@dlut.edu.cn; 3Key Laboratory of Ecophysics, College of Sciences, Shihezi University, Shihezi 832003, China

**Keywords:** dielectric barrier discharge plasma, heterogeneous catalyst, element doping, defect-rich, plasma catalysis

## Abstract

Dielectric barrier discharge plasma is one of the most popular methods to generate nanthermal plasma, which is made up of a host of high-energy electrons, free radicals, chemically active ions and excited species, so it has the property of being prone to chemical reactions. Due to these unique advantages, the plasma technology has been widely used in the catalytic fields. Compared with the conventional method, the heterogeneous catalyst prepared by plasma technology has good dispersion and smaller particle size, and its catalytic activity, selectivity and stability are significantly improved. In addition, the interaction between plasma and catalyst can achieve synergistic effects, so the catalytic effect is further improved. The review mainly introduces the characteristics of dielectric barrier discharge plasma, development trend and its recent advances in catalysis; then, we sum up the advantages of using plasma technology to prepare catalysts. At the same time, the synergistic effect of plasma technology combined with catalyst on methanation, CH_4_ reforming, NO_x_ decomposition, H_2_O_2_ synthesis, Fischer–Tropsch synthesis, volatile organic compounds removal, catalytic sterilization, wastewater treatment and degradation of pesticide residues are discussed. Finally, the properties of plasma in catalytic reaction are summarized, and the application prospect of plasma in the future catalytic field is prospected.

## 1. Introduction

The plasma followed by solids, liquids and gases was later referred to as an ionized gas of the “fourth state of matter”. In 1928, Langmuir first proposed the term “plasma” [1]. Different from the properties of the three substances above, plasma is made up of a host of high-energy electrons, free radicals, chemically active ions and excited states. The total electric charge of electrons and various negative ions is roughly equal to that of positive ions, so it is generally electrically neutral. In effect, plasma has been found in several applications like materials science and microelectronics industries, and even in many emerging environments such as medical. Plasma is divided into low and high temperature plasma on the basis of the temperature of internal electrons. Based on thermodynamic equilibrium, the low temperature plasma can be further divided into thermal plasma and non-thermal plasma (NTP). Dielectric barrier discharge (DBD) is a non-thermal plasma. After the thermal plasma reaches a local equilibrium state, the electron temperature is nearly close to the overall gas temperature [2]. This group mainly contains various plasma torches.

Compared with other common gases, plasma has the following characteristics: as a collection of charged particles, plasma has the conductivity similar to that of metal in which some ions collide, and the movement of positive and negative charges generates strong electric and magnetic fields; therefore, it has heat conduction and heat radiation. The total amount of positive and negative charges in the plasma is always equal. Any small space charge density changes will generate a huge electric field intensity, which will restore its original state and maintain its neutral state. The chemical character is lively, and it is prone to chemical reactions. The plasma processes have successfully been used in the two catalytic fields, namely catalyst preparation and direct excitation of reactants [3]. Catalysts play a pivotal role in the modern industry. Currently, nearly all chemicals in life are manufactured through catalytic processes. However, heterogeneous catalysis makes an important impact in these catalytic processes [4].

DBD research has now been going on for more than a century. Siemens et al. [5] focused on the production of ozone, and the first experimental report was made in 1857. The discharge experiment device they designed has many novel features, one of which is that the electrode does not contact with the plasma and is located outside the discharge chamber. Therefore, DBD has been considered to be “ozone discharge” for a long time. Andrew and Tait [6] called it “silent discharge” due to its quiet and silent discharge process in 1860. Buss [7] made a significant contribution to characterizing the discharge. He discovered that air breakdown between parallel electrodes covered by dielectrics is always accompanied by plenty of tiny short-lived current wires. DBD is one type of the non-thermal plasma (NTP), which can substitute the conventional catalytic chemical process that operates under high temperature conditions. Figure 1 represents two typical DBD electrode devices, namely planar and cylindrical DBD [8]. The placement of one or more dielectric layers between the metal electrodes is essential for the discharge operation. Typical dielectric materials are glass, quartz and ceramic.

In a DBD, the average temperature of high-energy electrons is very high, over the range of 10,000–100,000 K, but the actual gas temperature is close to the environment temperature. By electron impact ionization and excitation, source gases can produce active radicals, ions, excited atoms and molecular species [9]. The cold plasma characteristics of DBD, as well as the bombardment of high-energy electrons and other reactive ions generated by gas ionization, can achieve the effects that conventional means cannot achieve in the preparation and synthesis of catalysts. Compared with the control of a series of conditions such as temperature, pH and pressure and additives in liquid phase synthesis, the atmospheric pressure operation and simple structure of DBD make it widely used in the preparation of catalysts [10]. In addition, the plasma and catalyst exhibit some interdependence, which affect each other mutually [11].

Is plasma technology the solution to the problem? Despite extensive research in astrophysics and fusion plasmas has been conducted, the understanding of industrial plasmas is still limited [12]. In this work, we first introduce the properties, development trends and applications of the plasma in catalysis, and then focus on the advantages of DBD plasma preparation catalysts. The heterogeneous catalyst prepared by plasma has good dispersion and smaller particle size, and its catalytic activity, selectivity and stability are significantly improved. Secondly, we pay attention to the synergy of plasma and catalyst. Moreover, the interaction between plasma and catalyst can achieve synergistic effects, which significantly enhances the catalytic effect. Meanwhile, the application of plasma technology in the synergistic catalytic reaction such as methanation, CH_4_ reforming, NO_x_ decomposition, H_2_O_2_ synthesis, Fischer–Tropsch synthesis, volatile organic compounds removal, catalytic sterilization, wastewater treatment and degradation of pesticide residues is discussed. Finally, the properties of plasma in the catalytic reaction are summarized, and the application prospect of plasma in the future catalytic field is prospected.

## 2. Preparation of Catalyst by DBD Plasma

The core of catalysis mainly consists of three parts, namely chemical, chemical engineering and materials science. Due to the vigorous development of catalyst market, various types of catalyst publications are also increasing year by year. There are two types of catalysts currently in use: heterogeneous and homogeneous catalysts. Although the use of catalysts conforms to the principles of green chemistry, the preparation of the catalyst is not truly green. Innovative methods for preparing catalysts have been the focus in research. Preparation methods like microwave heating, plasma, biochemistry, etc. have been applied. Among these innovative methods, plasma has received extensive concern [13]. The plasma contains the active material prepared and processed by the catalyst. Under the action of plasma, there is a difference with conventional methods in the process of catalyst nucleation and crystal growth. Highly dispersed catalysts, defect-rich catalysts and heteroatom-doped catalysts can be prepared by plasma technology. In addition, plasma technology can also be widely used for etching, coating, and surface cleaning.

### 2.1. Highly Dispersed Catalyst

The purpose of catalyst preparation is for gaining products with high activity, selectivity and stability. The preparation of a large specific surface area catalyst requires deposition of the active metal component on the porous and thermostable material surface, which can not only enhance its thermal stability, but also improve the service life of the catalyst [14].

Researchers have applied methods such as impregnation and deposition–precipitation, etc. to prepare supported metal catalysts. These methods have significant effects on catalyst synthesis. However, they also have some disadvantages, like being time-consuming. Supported catalysts play an important role in energy and environment fields. A quick and easy way to prepare a supported metal catalyst is to combine DBD and cold plasma jets. Cold plasma preparation of supported metal catalysts mainly involves the reduction of metal ions. Depending on the types of metal element supported, it can be divided into precious metal supported catalysts and non-precious metal supported catalyst.

#### 2.1.1. Precious Metals

Supported precious metal catalysts are mainly applied in commercial processes such as automobile exhaust gas catalysis [15]. Over the years, Au, Ag, Pt, Pd or metal compounds have been widely used as industrial catalysts. One particular advantage of using supported precious metal catalysts is that the carrier can disperse the metal over a larger surface area.

Kim et al. [16] used atmospheric DBD plasma to assist reduction of supported platinum catalyst and cobalt catalyst. The reduction characteristics of prepared catalysts were tested. Di et al. [17] prepared different single metal (Pt [18], Ag [19], Pd [20]) supported TiO_2_ powders via DBD cold plasma. Moreover, they further prepared Pd/C [21], Pd/Al_2_O_3_ [22] and bimetallic Pd-Cu/Al_2_O_3_ [23,24] catalysts for CO oxidation. The photodegradation activity of methylene blue (MB) and CO oxidation activity were prominently improved. The schematic of atmospheric-pressure DBD cold plasma apparatus is demonstrated in Figure 2a. The X-ray diffraction (XRD) patterns of the activated carbon, Pd/C-PC (calcined at 300 °C in H_2_ for 2 h) and Pd/C-PW (washed with deionized water and then dried at 120 °C for 3h) were displayed in Figure 2b. It can be observed that all samples showed the carbon structure of the activated carbon at 25.0° and 43.5° broad diffraction peaks. The histograms of the particle size distributions of Pd nanoparticles for Pd/C-PC and Pd/C-PW correspond to Figure 2c,d, respectively. Pd nanoparticles were 1.92 nm and 15.93 nm in diameter, respectively. On the activated carbon surface for Pd/C-PC, smaller Pd nanoparticles are evenly distributed. Xu et al. [25] prepared graphene with good Pd dispersion through DBD cold plasma. It has been proved that the novel catalyst has high hydrodesulfurization catalytic activity. Jang et al. [26] treated Au/TiO_2_ catalysts by atmospheric-pressure O_2_ DBD plasma for CO oxidation. The sample treated by plasma (S-P) has higher activity and smaller particle size distribution than the calcined (S-C). Di et al. [27] prepared high CO oxidation activity Au/TiO_2_ catalysts by means of O_2_ DBD plasma. Moreover, since the compound of Au ion has been entirely reduced, the catalyst shows high CO oxidation activity.

#### 2.1.2. Non-Precious Metals

Due to the scarcity of precious metals and high prices, people are constantly researching and developing non-precious metals or low-content precious metal catalysts. The supported metal catalysts, represented by Ni, have been widely used in the field of methanation. Hu et al. [28] used DBD plasma to treat co-precipitated NiCO_3_-MgCO_3_ to obtain an Ni/MgO catalyst for CO_2_ reforming of CH_4_. The results indicate that the catalyst prepared via plasma has a larger specific surface area and a smaller particle size, and the CH_4_ and CO_2_ conversion rate is increased by more than 20%. Li et al. [29] studied the preparation of CO_2_ methanation reaction Ni/Ce/SBA-15-P catalyst through DBD plasma. From the N_2_ adsorption–desorption, XRD and TEM characterization results, it can be concluded that the Ni/Ce/SBA-15-P catalyst has an ordered mesoporous structure. Zhao et al. [30] used DBD combined with ammonia impregnation to prepare supported Ni catalysts. The prepared catalyst exhibits excellent activity, anti-coking property and anti-sintering performances in the methanation reaction of CO. Liu et al. [31] used DBD plasma to decompose a nickel nitrate to prepare the Ni/ZrO_2_ catalyst. The Ni particles are highly dispersed in the prepared catalyst and dramatically enhanced the methanation activity of CO. Li et al. [32] prepared an Ni/MgO catalyst for CO_2_ reforming of CH_4_ using DBD plasma. It was found that the plasma system can improve the catalytic property of catalysts. Liu et al. [33] prepared an Ni/MgAl_2_O_4_ catalyst by DBD decomposing Ni precursor for CO_2_ methanation. The CH_4_ yield of the catalyst prepared by plasma system is 71.8%, compared with 62.9% of the catalyst prepared by thermal at 300 °C. Yu et al. [34] used two-dimensional DBD plasma-treated vermiculite (2D-PVMT) to prepare an Ni-loaded catalyst system for CO methanation. The plasma treated catalyst exhibited superior performance and the CO conversion rate reached 93.5%. Liu et al. [35] decomposed an Ni precursor to prepare a Ni/SiO_2_ catalyst through DBD plasma and used it for CH_4_ steam reforming. The catalyst prepared this method exhibits intrinsically low activity and small size.

In addition to preparing catalysts supported Ni metal, catalysts supported other elements have been studied and applied in related fields with certain success. Ye et al. [36] found that the manganese oxide catalyst exposed to DBD plasma has lower particle size, higher dispersion and larger specific surface area. The quantity of specific sites (such as vacancies, angular atoms, edges, etc.) on the surface of the prepared catalysts increases, resulting in an increase in plasma reactivity. Li et al. [37] investigated how to use plasma precursor decomposition technology to prepare more efficient supported Co carbon nanotubes (Co/CNTs) catalyst. DBD plasma can prepare a Co/CNTs catalyst in which Co particles are evenly dispersed. Yu et al. [38] used two methods to prepare W_2_N catalysts for acetylene hydrochlorination. The first was prepared by depositing tungsten onto an activated carbon support, whereas the second was prepared from using the first catalyst via plasma treatment. It was found that plasma treatment increased the interaction between the support material and active components, and the catalyst activity increased by 12% compared with the untreated catalyst. El-Roz et al. [39] obtained TiO_2_-β zeolite composite catalyst via DBD plasma under O_2_ atmosphere. Under UV irradiation, the photooxidation rate of TiO_2_-β in methanol is eight times that of the conventional P25-TiO_2_ catalyst.

The catalysts synthesized via cold plasma typically have smaller metal nanoparticles and a high distribution, enhancing metal–carrier interactions. Hence, it usually has higher selectivity, stability and catalytic activity than catalysts synthesized by conventional means [40]. The method of preparing high dispersion catalyst by plasma system is shown in Table 1.

### 2.2. Defect-Rich Catalyst

Indeed, synthesizing a perfect crystal without any defects is basically impossible. There are many defects in nanomaterials that can regulate the electronic and surface properties of materials. It is well known that point, line, plane, and volume defects are found in materials. The point defect can be further grouped vacancies, interstitial atoms, impurities and heteroatom doping according to its composition and position [41]. There have been many reports that the surface defects of electrocatalysts may have a positive effect on the electrochemical reaction [42,43,44,45]. For instance, many researchers have reported that heteroatom-doped carbon materials possess better oxygen reduction reaction (ORR) activity compared to undoped pure carbon [46,47,48,49]. Therefore, there is a great research space for surface regulation of electrocatalysts around defect chemistry. The reason why we choose plasma technology to achieve the regulation of electrocatalyst surface structure is that plasma technology has the following functions: surface cleaning, oxygen cavitation, and etching.

#### 2.2.1. Carbon Material Defect

Huang et al. [50] studied the changes of surface morphology and chemical characteristics about carbon nanotubes after DBD modification. The plasma would strike the defect sites, so that the sp^2^-hybridized carbons of nanotubes is converted into sp^3^-hybridization, introducing COOH oxygen-containing functional groups. Meanwhile, the quantity of oxygen atom is reduced, which is due to the further oxidation of COOH group into CO_2_ and H_2_O at excessive exposure time. Paredes et al. [51] experimentally and theoretically investigated the properties and characteristics of atomic-scale defects on the surface of DBD plasma oxidized graphite. Two major types of defects can be observed by STM: protrusions with a diameter of ~1–5 nm and depressions of 5–7 nm width, the latter being a novel defect on the carbon surface.

#### 2.2.2. Metal and Metal Oxide Defect

Wang et al. [52] used a plasma etching technique to obtain a Co_3_O_4_-based OER electrocatalyst with high surface area and more oxygen vacancies. The Ar plasma etching can expose more surface sites. Plasma etching can adjust the electronic state by creating oxygen vacancies on Co_3_O_4_ nanosheets’ surface (Figure 3a). The oxygen vacancies produced by plasma etching were verified through XRD and XPS data plots. In order to prove that the plasma etched Co_3_O_4_ can expose more surface area, the N_2_ sorption isotherms of two nanosheets were obtained (Figure 3b). The results showed that the Brunauer–Emmett–Teller (BET) surface area of plasma etched Co_3_O_4_ nanosheets was much higher than that of the original Co_3_O_4_ nanosheets. The TEM images in Figure 3c,d show that the pristine Co_3_O_4_ nanosheets exhibit a continuous and compact surface. Figure 3e,f indicated the nanosheets have high oxygen evolution reaction (OER) activity after plasma etching. Cuenya et al. [53] used plasma treatments to prepare highly-defective nanostructured silver catalysts with selective CO_2_ reduction to CO under low overpotential. The results illustrated that the Ag catalyst prepared by plasma-treated produced small pore-like defects, and the activity and CO selectivity of the catalyst were greatly improved. Yu et al. [54] produced lots of oxygen vacancies and Ti^3+^ defects on the surface of TiO_2_ nanoparticles by Ar DBD plasma. Compared with pristine TiO_2_ nanoparticles, the energy band gap of Ar plasma treated TiO_2_ nanoparticles decreased from 3.21 eV to 3.17 eV, and the photocatalytic degradation ability of organic dyes was enhanced.

Plasma treatment can generate more active sites and high specific surface area deficient catalysts, which increases chemical reactivity. In addition, plasma technology can be used to detect defects. Ebihara et al. [55] applied DBD to detect defects in the surface of polytetrafluoroethylene coated metal. They developed a detection system for pinholes and cracks in Teflon film and used DBD technology to reduce residual material and damage to the film. The method of preparing a defect-rich catalyst using plasma is shown in Table 2.

### 2.3. Heteroatom-Doped Catalyst

Metals and metal oxides have a wide range of applications in many industrial processes, including synthetic materials, production and storage of clean energy. However, metal-based catalysts have many disadvantages, such as high cost, poor durability, low selectivity, easy poisoning, and environmental unfriendliness. Therefore, these shortcomings of precious metals hinder their large-scale commercial use in renewable energy [56]. Recently, heteroatom-doped catalyst materials are gradually entering the line of sight and exhibiting excellent performance in the catalytic process.

#### 2.3.1. Nitrogen Atom Doping

Lee et al. [57] used DBD pulsed laser deposition technique to grow N-doped ZnO thin films. Low temperature photoluminescence spectra show that N-doped ZnO film is p-type doping status. Yu et al. [58] used fast and effective DBD plasma treatment to obtain N-doped exfoliated graphene oxide (N-PEGO). N-PEGO becomes a potential cathode ORR catalyst in fuel cells due to its high nitrogen content, large specific surface area and enough active sites. Figure 4a displays the method of synthesizing N-PEGO. In short, N-PEGO was prepared by treating the ammonium carbonate dried GO dispersion with DBD plasma. Figure 4b,c are TEM images of N-PEGO. The TEM image shows that N-PEGO flakes have a relatively smooth layer structure. In the sample’s N 1s spectrum (Figure 4d), the four peaks centers of pyrrolic N, oxidic N species, pyridinic N and graphitic N are located at 400.3, 406.1, 399.0 and 401.4 eV, respectively, and the N content of N-PEGO is higher than that in GO by elemental analysis. As Figure 4e shows, linear sweep voltammetry (LSV) of GO, N-PEGO and 20 wt% Pt/C showed that the ORR onset potential of N-PEGO was 0.89 V, which was comparable to that of commercial 20 wt% Pt/C (0.93 V). Zhang et al. [59] prepared N-vacancy doped g-C_3_N_4_ with excellent photocatalytic performance through DBD plasma processing and applied it to the production of H_2_O_2_. As a result, it was found that the structure, morphology and optical properties of the catalyst were affected by the plasma treatment. Geng et al. [60] used wet impregnation to deposit CuO nanoparticles, and then processed by N_2_/Ar DBD plasma to obtain Cu_2_O/N-TiO_2_. Compared with CuO/TiO_2_ and TiO_2_, the photodegradation efficiency of methyl orange was prominently increased under visible light.

#### 2.3.2. Sulphur Atom Doping

Hu et al. [61] used DBD plasma to treat S-doped g-C_3_N_4_ catalyst with enhanced photocatalytic performance under H_2_S atmosphere. It was found that plasma treatment under H_2_S atmosphere can dope more S into the lattice of g-C_3_N_4_, which affected the morphology and electronic structure of the catalyst.

#### 2.3.3. Phosphorus Atom Doping

Wang et al. [62] prepared hydrogen evolution reaction (HER) and oxygen evolution reaction (OER) electrocatalyst by processing Co_3_O_4_ containing a P precursor with Ar plasma. The P atom can significantly tune the electronic structure and adsorption characteristics of Co_3_O_4_. The method of preparing heteroatom-doped catalyst using plasma is shown in Table 3.

## 3. DBD Plasma Catalysis

The synergistic effect means that the effect of combining the plasma with the catalyst is greater than the sum of their individual effects. Plasma catalysis synergy is a complex phenomenon in the interaction process between plasma and catalyst [63,64,65,66,67]. In general, the interaction of the plasma with the catalyst is divided into two parts. Firstly, plasma and catalyst independently affect the surface processes. Secondly, the plasma and the catalyst show some interdependence because the plasma interacts with the catalyst. This complex interdependence is graphically represented in Figure 5. It can be seen that the introduction of catalyst into the plasma discharge region may affect the discharge type and cause a change in electron distribution, which affects the generation of short-lived reactive plasma species.

### 3.1. Application of C1 Chemistry

The reaction in which the reactant contains only one carbon atom during the chemical reaction is collectively referred to as carbon-one chemical (C1). The subject of C1 is a compound containing only one carbon atom in the molecule, such as CO, CO_2_, CH_4_, CH_3_OH, etc. The main purpose of C1 is to save coal and oil resources, and generate more fuel with less carbon raw materials for humans. C1 products are a good substitute for petroleum. The combination of C1 chemistry and green chemical industry is to achieve coordinated development of economy and environment. No matter whether it is from the strategic perspective of world energy development, or focusing on environmental protection and sustainable development for the benefit of future generations, the combination of green chemical industry and C1 chemistry is of great significance. With the continuous progress of science and technology, a large number of novel and environment-friendly technologies have appeared in this field, and, in some industries, even traditional industrial methods have been replaced. With the progress of the times and the leap of science, green C1 chemistry will surely exert its great potential in the future chemical industry and better benefit mankind. We have previously summarized the application of two-dimensional layered double hydroxides in methanation and methane reforming reactions of C1 chemistry [68]. This paper mainly reviews the application of DBD plasma in C1 chemistry, such as CO methanation and oxidation, CO_2_ decomposition and CH_4_ reforming.

#### 3.1.1. Methanation

Nowadays, the gradual depletion of oil energy has become an urgent problem, which has increased the research on coal technology. Researchers have developed various technologies to convert coal into fuels such as methane and methanol [69]. The conversion of CO_2_ into CH_4_ is a valuable solution for carbon resource recycling in the C1 chemistry [70]. In the study of DBD plasma to convert CO_2_, the driving catalytic method combining plasma and catalyst is widely used. The advantage is that the heat generated in the plasma can make the catalyst work in the optimal temperature range, and the effect of the catalyst also allows the plasma to operate within the proper operating range.

CO_2_ hydromethanation is one of the important reactions in the field of energy and chemical industry. The principle of methanation reaction is that CO_2_ reacts with H_2_ under certain temperature and methanation catalyst to generate CH_4_ and steam. After cooling at the back, the water vapor is condensed and separated to obtain qualified hydrogen containing only CH_4_ impurities. Compared with other CO_2_ conversion methods, CO_2_ methanation has the advantages of fast reaction speed, high selectivity and few by-products. Simultaneously, the generated CH_4_ can be directly utilized, which effectively alleviates the problem of energy shortage. Industrially, the CO_2_ methanation reaction can also be used to prevent catalyst poisoning and crude hydrogen purification during ammonia synthesis. In addition, we have prepared different catalysts and applied them to the low-temperature CO methanation reaction, which has achieved good results [71,72,73,74]. Sabatier and Senderens first reported carbon monoxide methanation to methane in 1902 [75], which has been widely used in fuel cell industry and ammonia synthesis devices successively [76,77,78,79]. Ni catalyst is the most commonly used catalyst for methanation. Costa et al. [80] verified that DBD non-thermal plasma can improve the methanation performance of Ni/CeO_2_ and Ni/ZrO_2_ catalytic system, especially at lower temperatures. In the presence of plasma at 90 °C, CO_2_ conversion rate is up to 80%, while methane selectivity is 100%. In contrast, the same conversion rate and selectivity for the same catalyst requires a temperature of about 300 °C or higher without plasma. Jwa et al. [81] applied non-thermal DBD plasma for the heterogeneously catalyzed methanation of CO and CO_2_ over Ni-loaded alumina catalysts at 180–320 °C and atmospheric pressure. The results demonstrated that Ni-supported catalysts such as Ni/Al_2_O_3_ and Ni-TiO_2_/Al_2_O_3_ were significantly affected by plasma, which markedly increased the catalytic activity and led to the enhancement of methanation rate. Costa et al. [82] measured the activity of Ni-Ce-Zr hydrotalcite-derived catalysts in a mixed plasma catalyzed process to generate CH_4_ in low temperature. Even at very low temperatures, the methane yield is as high as 80%.

In addition, Cu, Mn and other noble metals like Ru and Rh also have excellent catalytic performance for methanation. Zeng et al. [83] reported the effect of γ-Al_2_O_3_ catalysts supported with various metals on the hydrogenation performance of CO_2_. The CO_2_ conversion was increased from 6.7% to 36% using both plasma and catalyst compared to plasma-catalyzed CO_2_ hydrogenation alone. Kim et al. [84] used DBD plasma-activated Ru/γ-Al_2_O_3_ catalyst to carry out CO_2_ methanation under an atmospheric environment. Regardless of the presence of catalyst, the CO_2_ conversion rate increased with DBD plasma. The addition of catalysts to the plasma may affect the discharge characteristics, thus producing highly reactive plasma materials. Figure 6a,b are diagrams of the experimental set-up for CO_2_ methanation and emission spectrum experimental device, respectively. Figure 6c illustrated the V–Q Lissajous figure of DBD plasma at 9 kV and 3 kHz on Ru/γ-Al_2_O_3_. Figure 6d presents a comparison of CH_4_ selectivity and CO_2_ conversion under DBD plasma conditions. In the absence of plasma, the CH_4_ selectivity and CO_2_ conversion are basically zero, and the Ru/γ-Al_2_O_3_ catalyst has no reactivity. In contrast, 8.21% CO_2_ conversion and 43.84% CO selectivity are obtained with plasma alone, but CH_4_ selectivity was reduced by 1.42%. When the DBD plasma was used together with Ru/γ-Al_2_O_3_ catalyst, CO selectivity was decreased, but the CO_2_ conversion and CH_4_ selectivity reach 12.80% and 73.30%, respectively. The influence of the H_2_/CO_2_ ratio on the CH_4_ selectivity and CO_2_ conversion is displayed in Figure 6e. The CO selectivity decreased as the H_2_/CO_2_ ratio increased; however, the CH_4_ selectivity and CO_2_ conversion increased. The effects of Ar on CO_2_ conversion, CH_4_ and CO selectivity are shown in Figure 6f. It can be seen that, when Ar is added, deoxygenation and methanation are simultaneously improved.

The results show that the introduction of catalyst into the plasma can vastly improve the conversion and selectivity of CO_2_ methanation reaction. Compared with the thermal reaction, the plasma reaction has a higher activity and can maintain the reaction at a lower temperature, avoiding the destruction of the catalyst surface properties by high temperature.

#### 3.1.2. CO Oxidation

CO oxidation has attracted much attention owing to its widespread application in automotive exhaust, indoor air and preferential CO oxidation of proton exchange membrane fuel cells [85,86,87,88]. Since the pioneering research of Haruta [89], supported gold nanometer catalysts have been considered as the most effective catalysts for CO oxidation at low temperatures. In the case of CO oxidation, it is speculated that the CO molecules may strongly adsorb on gold nanoparticles.

The supported gold nanocatalyst has an abnormally high CO oxidation activity at low temperatures, which can be widely applied to various industrial processes and instruments like environmental protection, clean energy conversion, CO sensors, air purification and gas mask. The plasma is treated under mild conditions without causing significant sintering of the calcined gold nanoparticles. Subrahmanyam et al. [90] conducted an experimental of CO monoxide oxidation by N_2_O in the DBD reactor. The direct oxidation of CO in the plasma reactor illustrated no apparent activity, but the presence of N_2_O increased the CO conversion. This is mainly attributed to the synergetic effect between plasma excitation and catalyst, which increases the performance of the reactor. Heintze et al. [91] investigated the DBD plasma-assisted partial oxidation of CH_4_ into synthesis gas. When Ni/Al_2_O_3_ catalyst is filled into the plasma discharge region, CO is oxidized to CO_2_ at 300 °C or higher, while the H_2_ and H_2_O selectivity keep stable.

#### 3.1.3. CO_2_ Decomposition

Fossil fuels such as coal and petroleum are the main energy sources for the development of modern industrial society, but the resulting CO_2_ gas has brought serious environmental and climate problems [92]. The use of fossil fuels produces CO_2_, and many studies have reported the impact of greenhouse gases on the climate. There has been a great interest in DBD plasma control pollution, which is now used in ozone production. Plasma-assisted catalytic techniques have been designed in conjunction with quartz barriers to decompose CO_2_. At the same time, some low carbon compounds such as CH_4_ were used in these studies [93].

In general, the performance of a DBD reactor depends on the configuration of the reactor, ambient gas, packed materials, gas flow rate and input power [94,95,96]. Subrahmanyam et al. [97] studied the effect of packing materials in a packed DBD plasma reactor on CO_2_ decomposition. The effects of porosity, dielectric constant and ultraviolet light on CO_2_ decomposition were studied. The study found that DBD plasma reactors will have improved performance in the case of various fillers. Uytdenhouwen et al. [98] studied the impact of gap size reduction and filler on CO_2_ conversion in the DBD microplasma reactor. Comparing the results with a conventional size reactor, although the energy efficiency is lower, decreasing the discharge gap can markedly enhance the CO_2_ splitting. These two improvements have been successful, and the combined effect of the two methods may be even better. Dai et al. [99] used DBD plasma in combination with the filler ZrO_2_ to study CO_2_ decomposition. It was found that the filler had a significant effect on the decomposition of CO_2_. When the catalyst is added, the CO selectivity was as high as 95% and the energy efficiency increased from 3.3% to 7%. Wang et al. [100] investigated the impact of filling materials on the CO_2_ conversion in the DBD microplasma reactor. The results show that the CO_2_ conversion in the dielectric filled reactor is higher than that in the without filling reactor. Tu et al. [101] performed coaxial DBD plasma photocatalytic reduction of CO_2_ to CO and O_2_ at low temperatures. In the case of only plasma, the maximum CO_2_ decomposition rate reached 21.7%. Figure 7a is a diagram of a reaction experimental device. The impact of BaTiO_3_ and TiO_2_ photocatalysts on CO_2_ conversion and energy efficiency is exhibited in Figure 7b. It is clear that the filling of TiO_2_ and BaTiO_3_ catalysts obviously increases the CO_2_ conversion and energy efficiency. Figure 7c presents the gas temperature and catalyst surface temperature in the DBD reactor. Obviously, in the first 15 min after the plasma is ignited, the plasma gas temperature without catalyst dramatically improved from 23.3 to 123.5 °C, after which it rises slowly and remains almost stable at 25 min. In the DBD reactor filled with BaTiO_3_ and TiO_2_, we noticed that the plasma temperature is almost the same as the catalyst surface temperature. Figure 7d indicates the reaction mechanisms of plasma-photocatalytic CO_2_ decomposition. The electron-hole pairs are produced by means of high-energy electrons in the gas discharge, which moves in the opposite direction under electric field, thereby decreasing the possibility of recombination. Ma et al. [102] increased the decomposition efficiency of CO_2_ by filling metal foam electrode in DBD plasma device. It was found that the metal foam not only functions as the energy transformation and electrode, but also consumed a part of the O_2_ and O radicals generated in the tube, and promoted the positive equilibrium shifting of the reaction.

#### 3.1.4. CH_4_ Reforming

The main component of natural gas is CH_4_, while CO_2_ is the final reaction product of carbon-containing compounds. Both of them are greenhouse gases and rich in carbon resources. Therefore, it is of great significance to develop the research on the basic organic chemical synthesis route of CH_4_ and CO_2_ as raw materials to replace coal and petroleum resources. However, due to their stable chemical properties, direct conversion requires extremely harsh reaction conditions, so plasma technology provides a new way for such conversion. The results demonstrate that the plasma reforming of CH_4_ and CO_2_ has the characteristics of large processing capacity, high conversion rate, high chemical energy efficiency and high calorific value yield. Simultaneously, CH_4_ and CO_2_ have mutual promotion effects in the reforming reaction.

Kogelschatz et al. [103] used DBD with zeolite catalyst to generate higher hydrocarbons directly from the CH_4_ and CO_2_ under ambient conditions, including alkanes, alkenes, oxygen-containing compounds and syngas. It can be seen that using the catalytic DBD plasma can achieve a cogeneration of syngas and higher hydrocarbons. Kraus et al. [104] conducted the combination of solid state catalysts with DBD for CO_2_ reforming of CH_4_, including CH_4_ and CO_2_ decomposition reactions. It is shown that the chemical properties of plasma can be improved by using catalytic coating in discharge. Cheng et al. [105] investigated the low temperature transformation of CO_2_ and CH_4_ in the coaxial DBD reactor. The experimental results show that the CO_2_ reforming of CH_4_ was improved by the introduction of steam through the synergistic catalysis of cold plasma and catalyst.

Furthermore, plasma technology can also be used for CH_4_ and steam reforming to produce H_2_. With global warming and the decline of fossil fuel, newly developed energy must satisfy both environmental and renewable requirements. H_2_ is one of the most promising energy sources in the future society, so it is very important to develop and implement systems for hydrogen production and storage.

Steam reforming of CH_4_ typically requires additional steam supply to control carbon formation. Catalysts with strong anti-coke properties help to decrease the amount of steam and save resources. Nozaki et al. [106] demonstrated the synergistic effect between plasma and catalyst by using Ni/SiO_2_ catalyst for plasma CH_4_ steam reforming. Figure 8a represents a schematic diagram of different electronic and kinetic processes. The electron collision ionization, excitation and dissociation of CH_4_ were evaluated with electron density and field intensity. Figure 8b shows the CH_4_ conversion curve for the DBD catalytic packed-bed reactor. The combination of Ni catalyst and barrier discharge significantly enhances product conversion. Figure 8c makes a comparison of three different reaction systems. It can observe a strong synergistic effect between the Ni catalyst and barrier discharge, and the CH_4_ conversion rate exceeded the sum of Ni catalyst and barrier discharge used alone. Figure 8d represents enthalpy balance of the typical reforming device. Figure 8e shows a potential curve of CH_4_. As shown in the figure, the plasma-enhanced catalytic reaction is interpreted as a two-step excitation. Kim et al. [107] realized the complete oxidation of CH_4_ by hybridizing both catalyst and plasma in a DBD quartz tube reactor. Under the combined action of the catalyst and the plasma, CH_4_ was oxidized at room temperature to generate CO_2_ with lower CO selectivity, showing complete oxidation with the assistance of catalyst. Nozaki et al. [108] described the steam reforming of CH_4_ in a plasma-catalyst hybrid reactor. It could observe a synergistic effect between Ni catalyst and the non-thermal DBD at lower reaction temperatures, and the CH_4_ conversion far exceeded equilibrium conversion; meanwhile, the selectivity of product was prone to balance composition under given conditions.

### 3.2. NO_x_ Decomposition

Since the 21st century, with the increase in the number of automobiles and the industry development, exhaust pollutants, especially nitrogen oxides (NO_x_), emissions have brought a great load to the environment, which causes serious harm to human health and attracts the general attention of the world’s research. Plasma catalysis has made great progress in academic research and industrial applications, which is a new processing technology with very optimistic prospects.

In the past decade, many researchers concentrate on technologies that reduce emissions or control air pollution. Among these solutions, plasma is preferred because of low energy consumption and high efficiency. The non-thermal plasma low-temperature catalytic conversion of NO_x_ to valuable chemicals or harmless gases has been effectively done [109]. Our research group also conducted experiments and reviews on selective reduction and denitrification of MnO_x_-Fe_2_O_3_/vermiculite monolithic honeycomb catalysts and porous microspherical aggregates of Mn–Ce–Fe–Ti mixed oxide nanoparticles at low temperatures [110,111,112,113]. Yu et al. [109] used non-thermal DBD plasma and H_2_O as external electrode to efficiently remove NO_x_. As illustrated in Figure 9, Yu’s team studied the change in NO conversion with and without glass beads using copper foil and H_2_O as external electrodes, respectively. As can be seen from the picture, when the copper foil is used as the external electrode, the NO conversion rate is 14.1%; however, when H_2_O is the external electrode, the NO conversion rate reaches 28.8%. Meanwhile, when the glass beads are packed with H_2_O as the external electrode, the NO conversion rate is greatly increased to 95.9%, indicating that the glass beads could greatly improve the NO removal rate. Niu et al. [114] investigated the synergistic effect of DBD plasma and Ag/Al_2_O_3_ catalytic system on C_2_H_2_ selective catalytic reduction of NO_x_. A significant synergistic effect was obviously observed in plasma and catalytic system. Wang et al. [115] used DBD plasma to assist with catalytic reduction of NO_x_ on the Mn–Cu catalyst. DBD combined with Mn-Cu catalyst improved the catalytic activity of selective catalytic reduction reaction and provided a new method for selective catalytic reduction research. Niu et al. [116] reported the synergistic effects of HZSM-5 catalyst modified by indium and pulsed DC DBD plasma on the selective reduction of NO_x_ by C_2_H_2_ at 200 °C.

### 3.3. H_2_O_2_ Synthesis

Hydrogen peroxide (H_2_O_2_) is an important green oxidant, which is widely used in daily life. At present, H_2_O_2_ is almost entirely produced by indirect and ungreen anthraquinone process (AQ). Some reports have investigated the direct synthesis of H_2_O_2_ through electrochemical devices and noble metal catalysts using H_2_ and O_2_ as raw materials. However, it is difficult to obtain high-purity H_2_O_2_ products without purification.

The semiconductor industry, including microelectronics, displays and photovoltaics, requires electronic grade H_2_O_2_. This high purity H_2_O_2_ must comply with the semiconductor equipment. Manufacturing high purity H_2_O_2_ from the commercial grade H_2_O_2_ of the AQ process requires complicated and energy-intensive purification techniques. Purification is dangerous due to the reactivity of H_2_O_2_. At the moment, reverse osmosis is considered as an ultra-purification method for H_2_O_2_, but there are no longer-life membrane materials. Therefore, it is critical to study a new technology for synthesizing high purity H_2_O_2_ [117]. Figure 10 shows the experimental device and reaction mechanism diagram of H_2_O_2_. It can be seen from the figure that, during the continuous operation, the O_2_ conversion rate remains stable, and the volume of the H_2_O_2_ product increases linearly with time.

Studies in the 1960s showed that H_2_O_2_ can be produced by free radical reaction in the presence of any catalyst or chemical catalyst in a H_2_/O_2_ non-equilibrium plasma. However, the plasma method does not attract much attention due to the low H_2_O_2_ yield and the safety of the H_2_/O_2_ reaction caused by discharge. To prevent fire and explosion, the O_2_ content must be strictly kept below 4 mol%.

Li et al. [118] studied the synergistic effect of TiO_2_ photocatalysis and plasma discharge on the degradation of thiamethoxam. The results indicated that the synergy effect can obtain high concentration of H_2_O_2_. Bruggeman et al. [119] reported a detailed study of the plasma morphology and H_2_O_2_ production in DBD operating at various powers and water vapor concentrations. They evaluated the effect of discharge morphology and power on the concentration dependence of the OH and H_2_O_2_ production through a joint study. Vasko et al. [120] investigated the H_2_O_2_ production by RF glow discharge in helium–water vapor mixtures. The consistency between the experiment and model is very well, which corresponds to the uncertainty of reaction rate and experiment accuracy to some extent. Hu et al. [121] used DBD plasma to synthesize N vacancies doped g-C_3_N_4_ catalyst under H_2_ atmosphere. The impact of doping N vacancies into g-C_3_N_4_ on the photocatalytic H_2_O_2_ production capacity was investigated. Guo et al. [122] developed a plasma reactor with multiple parallel DBD tubes to synthesize H_2_O_2_ from H_2_/O_2_. Thevenet et al. [123] synthesized H_2_O_2_ using DBD related to fibrous materials. In this work, H_2_O_2_ is synthesized by igniting a mixture of H_2_ and O_2_ in a DBD. This study demonstrates that non-thermal plasma can effectively synthesize H_2_O_2_, and the use of fibrous materials prominently enhances the synthesis process.

Research shows that the configuration of plasma reactor plays an important role in H_2_O_2_ synthesis. In a single DBD plasma reactor with a bare metal high voltage electrode and H_2_O ground electrode, the O_2_ conversion rate of H_2_/O_2_ mixture containing 3% O_2_ reached 100%, but the H_2_O_2_ selectivity was only 3.5%. However, 57.8% O_2_ conversion and 56.3% H_2_O_2_ selectivity can be obtained by using a dual DBD plasma reactor with a pyrex-covered metal high voltage electrode. Although selectivity has been vastly enhanced, low efficiency and safety remain a major challenge due to low O_2_ content. Even though H_2_O_2_ is less selective than the AQ process, for the direct production of high purity H_2_O_2_, simple plasma methods are attractive [124].

### 3.4. Fischer–Tropsch Synthesis

Fischer–Tropsch synthesis (FTs) is a technique that converts CO_2_, CH_4_ and waste biomass into value-added chemicals and fuels, which uses syngas to convert to alkenes, long-chain alkanes, oxygenates, alkenes and water [125]. With the depletion of petroleum resources and the deteriorating climate, FT synthesis will play an increasingly important role in solving energy crisis and developing low-carbon energy.

FT synthesis takes place at the Co metal site, and the total number of Co metal sites on the supported catalyst depends on the dispersion and reductibility of Co. Previous research has found that glow discharge plasma can significantly improve the Co dispersion, and the catalyst assisted by plasma jets exhibit higher Co dispersion. Plasma jet is a promising tool for controlling Co dispersion and improving the catalytic performance of cobalt FT catalyst [126].

Li et al. [127] prepared a zirconium modified Co/SiO_2_ FT catalyst by using DBD plasma. The results showed that, compared with those treated by the calcining method, the catalyst processed by DBD plasma demonstrated higher FT activity and heavy hydrocarbon yield. Akay et al. [128] investigated CO and H_2_ to higher hydrocarbons on Cu/Co catalyst prepared by DBD for FT synthesis. It is found that Cu/Co catalyst can improve the FT synthesis under low temperature and ambient pressure. Xu et al. [129] used DBD plasma to catalyze the conversion of CH_4_ to advanced hydrocarbons. The experiment has confirmed that DBD plasmas produce a higher CH_4_ conversion rate in the presence of CO_2_. Using CO_2_ as a coreactant obviously enhances the CH_4_ conversion rate and inhibits the formation of carbon deposits, Saleem et al. [130] used plasma-assisted decomposition of a biomass gasification tar analogue (toluene) to decompose them into lower hydrocarbons via DBD reactor. The toluene removal rate was reached 99% at a plasma power of 40 W and a residence time of 2.82 s. Figure 11a is a diagram of the experimental device. Non-thermal plasma is generated in a cylindrical DBD, as can be seen from Figure 11b. The aromatic C–C bonds is not enough to be destroyed at low power. While at high power, the average electron energy is high, and a larger proportion of lower hydrocarbons can be observed. Figure 11c shows that the yield of CH_4_ and the selectivity of C_2_-C_5_ products as a function of residence time, which increase with residence time. This may be attributed to the collisions between tar analogue and reactive substances leading to the cleavage of aromatic rings. The influence of power and temperature on toluene conversion is displayed in Figure 11d. It can be seen that, when the power is raised in the range of 20–40 W, the toluene decomposition remains unchanged. However, it ranges from 82% to 91% at 10 W. It may be reduced owing to the formation of solid deposits. Figure 11e shows the reaction mechanism diagram at high temperature. The benzene ring fragmentation increases methane production due to high energy electrons and excitations at high power.

### 3.5. Volatile Organic Compounds Removal

Volatile organic compounds (VOCs) are common atmospheric pollutants, mainly those with volatile organic compounds having a boiling point of 250 °C or lower. Aromatic, alcohols, ketones and esters are typical VOCs emitted from various human activities, like printing, painting, coil coatings, wood processing, etc. In addition, the most important and common consequence of emitting VOCs into the atmosphere is that they can cause stratospheric O_3_ consumption and tropospheric O_3_ formation. VOCs are important precursors of atmospheric pollutants like O_3_ and secondary organic particles. They can participate in photochemical reactions to produce secondary organic aerosols, which have an important impact on atmospheric visibility and global radiation balance, although some VOCs may in the short term will not cause serious harm to human health, but long-term exposure may cause mutagenic or carcinogenic effects [131].

Extensive research has been conducted to design low cost and effective VOC processing methods. Recently, several techniques for decomposing these organic contaminants through oxidation have been developed, especially the photocatalytic process using ultraviolet irradiation, in which HO free radicals are produced to oxidize these harmful VOCs. In recent years, an effective method for non-thermal plasma oxidation of VOCs has been reported. It has been reported that placing catalyst in the discharge area can improve the efficiency of VOCs removal and CO_2_ formation [132].

Tatibouet et al. [132] observed synergetic effects by coupling photocatalysis with DBD plasma to remove low concentrations of isovaleraldehyde from air. The results showed that the plasma photocatalytic system obviously improved the isovaleraldehyde removal efficiency. Rtimi et al. [133] used DBD and photocatalysis in a continuous reactor to degrade butyraldehyde and dimethyl disulfide. A synergistic effect was observed with the butyraldehyde removal, but no synergistic effect was observed with dimethyl disulfide removal because of catalyst poisoning. Lu et al. [134] reported the removal efficiency of VOCs in non-thermal plasma double DBD, and also investigated plasma-catalyst synergistic effects on VOC removal. Bouzaza et al. [135] studied DBD plasma/photocatalysis combination, photocatalysis and DBD plasma to remove isovaleraldehyde from air, and the former produced a synergistic effect. Li et al. [136] investigated the synergistic effect of non-thermal DBD and catalyst for oxidation removal of toluene. The results demonstrated that the targets increasing energy efficiency and reducing O_3_ in exhaust gas are achieved. Jo et al. [137] used a DBD reactor to decompose VOCs and found that humidity was a limiting factor in the non-thermal decomposition of VOCs. It was found that the optimum removal rate was 20% relative humidity. The graphical abstract is illustrated in Figure 12a. The impact of relative humidity on decomposition efficiency at different frequencies is displayed in Figure 12b. It can be seen from the figure that, when the frequency keeps constant, the decomposition efficiency first increases and then decreases with the relative humidity; when the relative humidity is constant, the higher the frequency, the greater the decomposition efficiency. Figure 12c presents the Lissajous figures of various discharge regions, which depends on the amount of transfer charge. Figure 12d is the impact of specific energy density on CO_2_ selectivity and decomposition efficiency. Although higher SED results in higher decomposition efficiency, it does not exhibit any consistent linear relationship.

### 3.6. Catalytic Sterilization

The indoor environment in which humans are often exposed to various types of pollution has already affected all aspects of our lives, especially organic pollution and bacterial pollution, which have severely restricted the improvement of quality of life. The existing filtration, adsorption, negative oxygen ions and ozone technology are difficult to continuously and effectively purify organic pollution, and disinfectant, ultraviolet lamp sterilization, ozone sterilization and other technologies have problems such as side effects on the human body and low sterilization efficiency. The high oxidation performance of photocatalytic technology has a strong bactericidal property, which can kill bacteria and viruses by destroying the cell wall of bacteria and coagulating proteins of viruses. Compared with photocatalyst, plasma discharge catalysis can produce more hydroxyl radicals, which can decompose harmful substances and bacteria viruses in the air more quickly and effectively.

Choi et al. [138] analyzed the sterilization effect by pulsed DBD. The experimental results showed that, when the DBD treatment time was 70 s, 99.99% *Escherichia coli* was sterilized, and the O_3_ molecule is the main bactericidal species. Kostov et al. [139] performed bacterial sterilization by air DBD, and all bacterial cells were killed by DBD treatment for 20 min. Yi et al. [140] conducted DBD sterilization experiments on *Escherichia coli* and *Bacillus subtilis* in drinking water. The results showed that, with the increase of input voltage V and reaction time t, the sterilization rate increased significantly. The optimum sterilization effect was achieved at a pH of 7.1. Nagatsu et al. [141] studied a flexible sheet-type DBD that compared the low temperature sterilization of the wrapped material by adjusting the N_2_ and O_2_ ratios. Kikuchi et al. [142] studied the effects of environmental humidity and temperature on sterilization efficiency of atmospheric pressure DBD plasmas. The results showed that the inactivation of bacteria was greatly affected by humidity. Roy et al. [143] examined the role of O_3_ in surface plasma sterilization by using DBD plasma. The results demonstrate that O_3_ plays an important role in the plasma sterilization process, and the energy flux of electrode also plays a crucial role in plasma sterilization. Hong et al. [144] studied a novel multihole DBD (MH-DBD) plasma sterilization system for the sterilization of aqua pathogens. It was found that the MH-DBD plasma exhibited a higher sterilization performance over 6.5–7.5 log reduction. As is exhibited in the Figure 13a, the electric field is distributed symmetrically along the high voltage electrode in the coaxial DBD system. However, the MH-DBD system (Figure 13b) only shows asymmetric distribution of a strong electric field near the hole. Figure 13c displays the kill curve of Vibrio harveyi 12724. It can be observed that the presence of nitrogen in the DBD plasma system does not have a bactericidal effect on the microorganisms. In contrast, the nitrogen in the MH-DBD plasma system was continuously sterilized during the 5 min treatment period. As is demonstrated in Figure 13d, the coaxial DBD and MH-DBD plasma systems were continuously sterilized for 5 min. Figure 13e indicates the optical emission spectrum, which helps to identify the various excitations produced by the MH-DBD plasma. Figure 13f shows the kill curves of the pathogens processed by the MH-DBD plasma. Since the MH-DBD plasma can generate stable, sustainable oxidant, this remarkable sterilization effect was maintained for 4–5 days, after which the pathogen gradually proliferated due to its growth rate.

### 3.7. Wastewater Treatment

Thanks to the low removal rate of many harmful substances in wastewater, there are large amounts of man-made compounds in urban, agricultural and industrial wastewater, which can enter the natural surface and underground water bodies. Some substances owing to persistence get great attention and toxic effects after release into the receiving system [145]. The presence of hazardous chemicals in natural waters is a very serious issue. To reduce harmful effects of these harmful substances on humans and environment, it is necessary to find suitable and economical techniques to reduce their concentration.

Plasma treatment has been used to experimentally remove various substances from water, including cyanide, VOCs, phenols, organic dyes and drugs. However, there are few guidelines to remove more refractory compounds and their energy efficiency and degradation kinetics in wastewater. It is well known that plasma treatment can produce degradation by-products, so we must conduct a detailed study.

#### 3.7.1. Pharmaceutical Wastewater

Fang et al. [146] reported the use of DBD to degrade aniline wastewater. The results show that the addition of a certain amount of Na_2_CO_3_ and H_2_O_2_ to the wastewater can promote the degradation of aniline.

#### 3.7.2. Dyestuff Wastewater

Mok et al. [147] studied the application of O_3_ and ultraviolet light generated by DBD reactor for the oxidative degradation of organic contaminants in an azo dye Acid Red 27 wastewater. It was found that the ultraviolet light alone can degrade quite a large number of organic pollutants. The ozonation also had a great influence on the degradation of organic pollutants, and when ozonation and photocatalytic effects were combined, the degradation effect was remarkably improved. Zhong et al. [148] used ns-pulse DBD plasma to degrade dye wastewater. Experiments indicate that the gas phase non-equilibrium plasma generated by pulse power can effectively decompose indigo carmine in atomized aqueous solution. Sun et al. [149] reported the study of ozonation and DBD plasma induced photocatalysis treatment of azo dye Acid Red 4 wastewater. The results indicate that the system has the ability of photocatalysis and ozonation to degrade organic contaminants, which is a promising wastewater processing technology. Attri et al. [150] investigated the impact of γ-ray and DBD plasma treatments on dye-polluted water treatment. The wastewater treatment efficiency at different time intervals was studied. Martuzevicius et al. [151] examined the degradation of various textile dyes by semi-continuous DBD plasma. It was found that plasma treatment can reduce the toxicity of wastewater to near zero.

#### 3.7.3. Grease Wastewater

Kuraica et al. [152] studied the application of coaxial DBD for potable and oil derivative wastewater treatment, which obviously reduced the chemical oxygen demand and potassium permanganate demand in wastewater treatment.

#### 3.7.4. Antibiotic Wastewater

Yuan et al. [153] added persulfate (PS) to DBD plasma device to generate and activate SO_4_^2−^ groups, and determined the effect of PS addition and applied voltage on tetracycline (TC) removal efficiency. The results illustrated that the addition of PS had a prominent synergistic effect, which promoted the removal rate and degradation efficiency of TC. Figure 14a shows a gas phase surface discharge device diagram. The discharge power is calculated according to the Lissajous figure (Figure 14b). Figure 14c indicates the effect of PS addition ratio on TC removal. It was concluded that, when the molar ratio increased from 5:1 to 120:1, the TC removal rate was dramatically improved due to the addition of PS. When the molar ratio was higher than 20:1, the efficiency was no longer increased and actually decreased slightly. Figure 14d,e show the influence of ethanol on the removal of TC and H_2_O_2_ production. Clearly, adding the ethanol inhibited the removal of TC in the absence or presence of PS. After 15 min of treatment, the removal efficiencies of the DBD and DBD + PS systems alone reduced from 87.5% and 83.2% to 82.6% and 70.1%, respectively. The H_2_O_2_ concentration could be used as a characterization of the oxidation capability. As can be seen from the figure, compared with no PS, the H_2_O_2_ concentration was higher when PS was added, and the H_2_O_2_ concentration in deionized water was higher than TC solution. The total ion chromatogram and HPLC/MS chromatograph of the TC degradation intermediates were depicted in Figure 14f. The TC was clearly removed after 15 min processing compared to the untreated sample TC peak intensity.

### 3.8. Degradation of Pesticide Residues

Pesticides are a large variety of different types of compounds that are used to delay crop spoilage and prevent pests. However, as pesticide resistance increases, their residues on food and their long-lasting effects have attracted widespread attention. Agricultural chemicals like pesticide residues on a fresh product can pose serious health risks if not washed. The possibility of pesticides’ degradation in food and water by plasma discharge has recently been reported. Low-temperature plasma has become a potential biological purification technology, which is used to treat microorganisms in food like fruits and vegetables and reduce chemical risks.

#### 3.8.1. Insecticide

Cullen et al. [154] selected three experimental pesticides, dichlorvos, malathion, endosulfan, and used atmospheric air high voltage DBD plasma reactor to degrade pesticides in water. After plasma treatment for 8 min at 80 kV, the degradation efficiency of pesticides were found to be 78.98% for dichlorvos, 69.62% for malathion and 57.71% for endosulfan, respectively. Shi et al. [155] processed spinach and apple samples contaminated with omethoate using low temperature plasma (LTP) generated by dielectric barrier corona discharge. It is found that omethoate residue in vegetables and fruits could be effectively degraded with proper dosage of LTP without affecting the quality of fruits and vegetables. Bai et al. [156] reported the degradation of dimethoate induced by DBD plasma in aqueous solution. The impacts of degradation pathway and parameters on dimethoate solution were studied. Li et al. [157] studied the degradation of acetamiprid in wastewater in DBD reactor. It can be concluded that acetamiprid could be effectively removed from aqueous solution and hydroxyl radicals played an important role in the degradation process.

#### 3.8.2. Herbicide

Zheng et al. [158] reported the research on diuron degradation with DBD plasma and proposed the degradation mechanism. Roglic et al. [159] investigated the influence of different catalysts on mesotrione degradation in water falling film DBD reactor. Wardenier et al. [160] studied an innovative advanced oxidation processes based on a continuous-flow pulsed DBD reactor discharge.

A synthetic micropollutantsmixture containing five pesticides, two pharmaceuticals, and one plasticizer were used. The results showed that the total removal efficiency of all studied micropollutants was greater than 93.8%, and the energy efficiency varied between 2.42 and 4.25 kWh/m^3^. Valsero et al. [161] used DBD atmospheric plasma to remove pollutants from wastewater containing atrazine. However, when the solution to be processed contains high concentrations of organic substances and mineral salts, the efficiency is decreased, which may inhibit or compete with oxidant substances generated by plasma.

#### 3.8.3. Germicide

Cullen et al. [162] reported the degradation of pesticide residues on strawberries by DBD nonthermal plasma technology. After 80 kV treatment for 5 min, the contents of azoxystrobin and fludioxonil decreased by 69% and 71%, respectively. Figure 15a presents the diagram of the DBD experimental device for NTP treatments inside the package. Figure 15b shows the emission spectrums with or without strawberries. The emissions were stronger without any samples in the package. Figure 15c representative chromatogram showing retention times of the four pesticides. As can be seen from the figure, the retention time of the four pesticides, namely azoxystrobin, cyprodinil, fludioxonil and pyriproxyfen were 36.6, 20.6, 24.1 and 29.8 min, respectively. Figure 15d,e indicate the residual pesticide concentrations in strawberries before and after plasma treatment. The pesticide residue concentration of all samples was significantly reduced compared with the control.

## 4. Conclusions

As an emerging frontier hotspot technology, plasma has been applied in many research fields. In this review, we first introduce the characteristics of plasma, such as conductivity, heat conduction, heat radiation, active chemical properties, easy to produce chemical reactions and its recent advances in catalysis. The advantages of preparing various catalysts by DBD plasma technology are summarized, including highly dispersed catalyst, defect-rich catalyst and heteroatom-doped catalyst. Compared with the catalyst prepared by the conventional method, the heterogeneous catalyst prepared by plasma exhibits good dispersion and smaller particle size, and its catalytic activity, selectivity and stability are significantly improved. Meanwhile, the chemical reaction activity is remarkably enhanced, attracting more and more attention from all walks of life.

Furthermore, we explored plasma catalysis, which can dramatically improve the catalytic reaction effect. For example, plasma technology can be used to generate clean fuel CH_4_ via CO_2_ as a raw material; under the action of an external electric field, complex macromolecular pollutants can be converted into small molecules safe and free of pollutants; plasma flue gas denitration is realized by combining electron beam technology with pulse corona discharge technology; meanwhile, plasma technology can also purify automobile exhaust and dust. Finally, the properties of plasma in the catalytic reaction are summarized, and the application prospect of plasma in the future catalytic field is prospected. At present, scientists have applied the plasma technology in the latest research fields like the “artificial sun”, aerospace, stomatology, photonic crystal and spectroscopic optics, which has achieved certain results. With the development of plasma technology, low-temperature plasma will definitely be applied in a wider range and more industries, and the study on low-temperature plasma will also be more in-depth.

## Figures and Tables

**Figure 1 nanomaterials-09-01428-f001:**
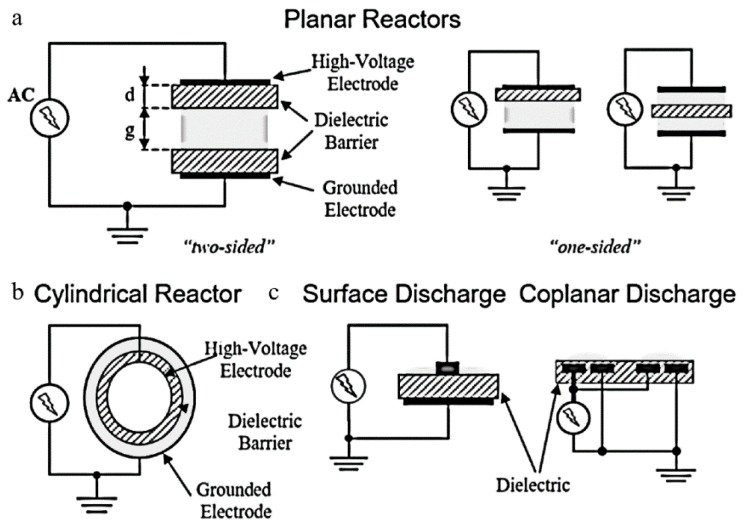
Two typical DBD electrode devices: (**a**) planar reactor; (**b**) cylindrical reactor; (**c**) two discharge types of surface discharge and coplanar discharge. Reproduced with permission from [8]; Copyright Elsevier, 2003.

**Figure 2 nanomaterials-09-01428-f002:**
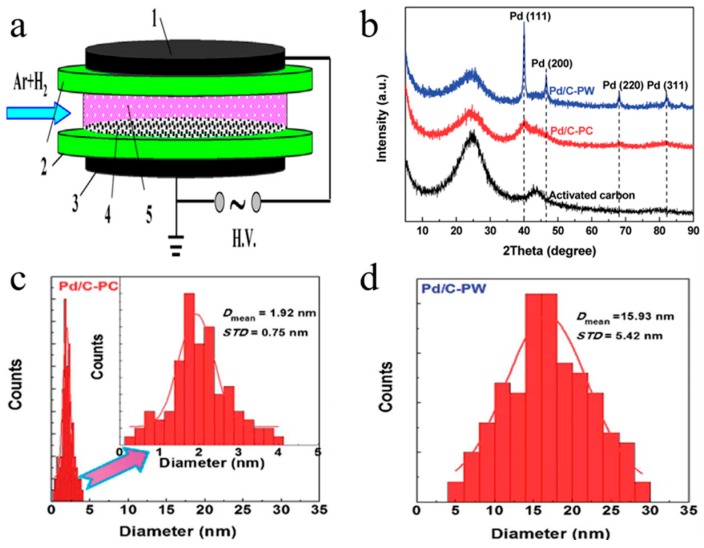
(**a**) the schematic of DBD cold plasma apparatus (1—high voltage electrode, 2—quartz glass, 3—low voltage electrode, 4—sample, and 5—cold plasma); (**b**) XRD patterns of the Pd/C-PW, Pd/C-PC and activated carbon; (**c**) the histograms of the particle size distributions of Pd nanoparticles corresponding to Pd/C-PC; (**d**) the histograms of the particle size distributions of Pd nanoparticles corresponding to Pd/C-PW. Reproduced with permission from [17,21]; Copyright Elsevier, 2013 and Royal Society of Chemistry, 2014.

**Figure 3 nanomaterials-09-01428-f003:**
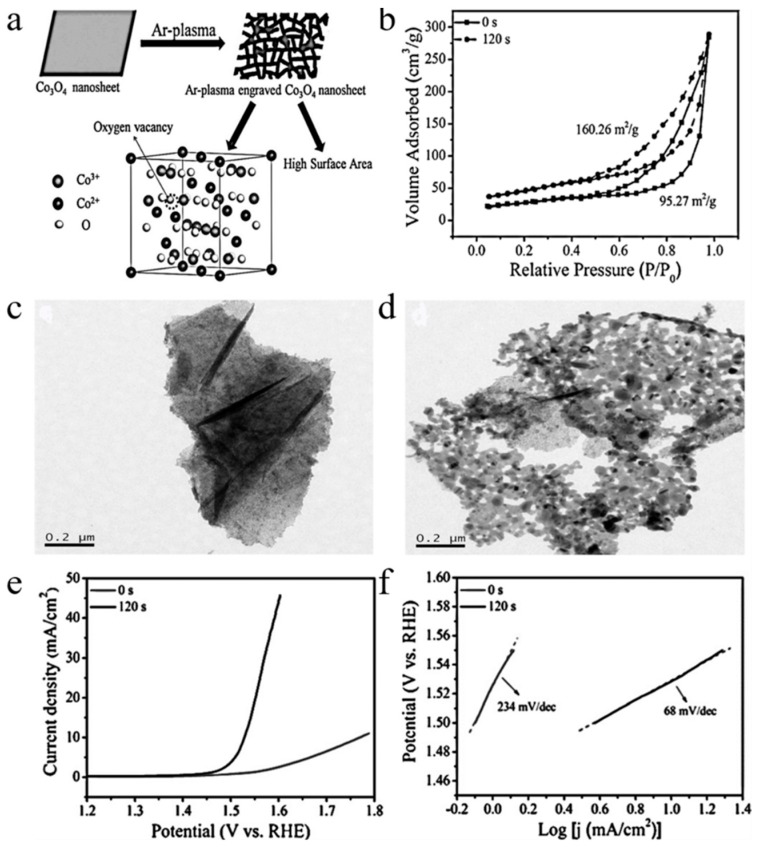
(**a**) graphical illustration of Ar plasma etching of Co_3_O_4_; (**b**) N_2_ sorption isotherms of plasma-etched and original Co_3_O_4_; (**c**) TEM images of original Co_3_O_4_; (**d**) TEM images of Ar plasma etched Co_3_O_4_; (**e**) the polarization curves of OER on original Co_3_O_4_ (0 s) and the plasma etched Co_3_O_4_ (120 s); (**f**) Tafel plots. Reproduced with permission from [52]; Copyright Wiley, 2016.

**Figure 4 nanomaterials-09-01428-f004:**
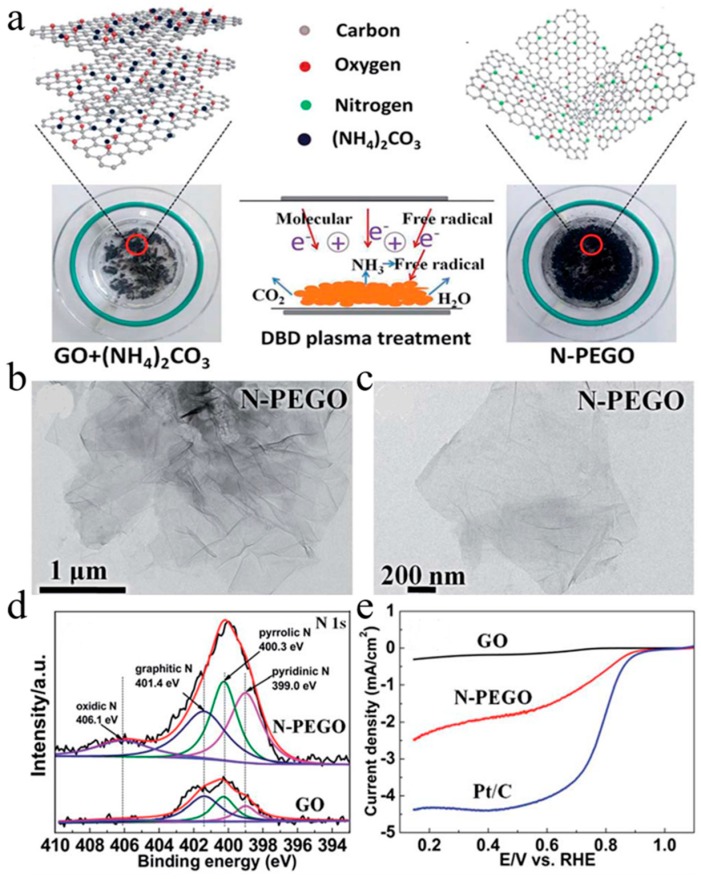
(**a**) the method of synthesizing N-PEGO; (**b**,**c**) TEM images of N-PEGO; (**d**) the N 1s spectrum of the samples; (**e**) LSV curves of GO, N-PEGO and 20 wt% Pt/C. Reproduced with permission from [58]; Copyright Royal Society of Chemistry, 2018.

**Figure 5 nanomaterials-09-01428-f005:**
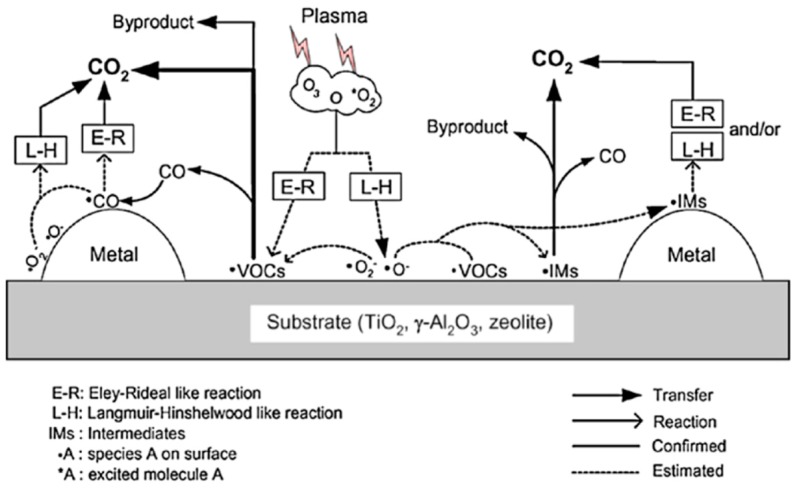
Complex interdependence of plasma–catalyst interactions. Reproduced with permission from [11]; Copyright American Chemical Society, 2015.

**Figure 6 nanomaterials-09-01428-f006:**
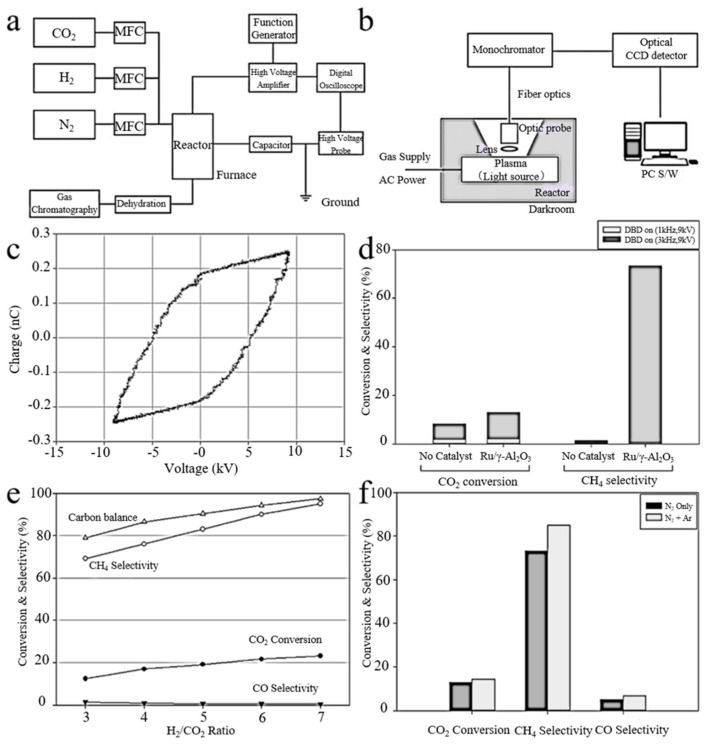
(**a**) the diagram of a methanation experimental device; (**b**) the diagram of experimental device for optical emission spectroscopy; (**c**) Lissajous diagram of the DBD plasma; (**d**) comparison of the CH_4_ selectivity and CO_2_ conversion; (**e**) effect of the H_2_/CO_2_ ratio on the CH_4_ selectivity and CO_2_ conversion for Ru/γ-Al_2_O_3_; (**f**) effect of Ar on DBD over Ru/γ-Al_2_O_3_. Reproduced with permission from [84]; Copyright Elsevier, 2017.

**Figure 7 nanomaterials-09-01428-f007:**
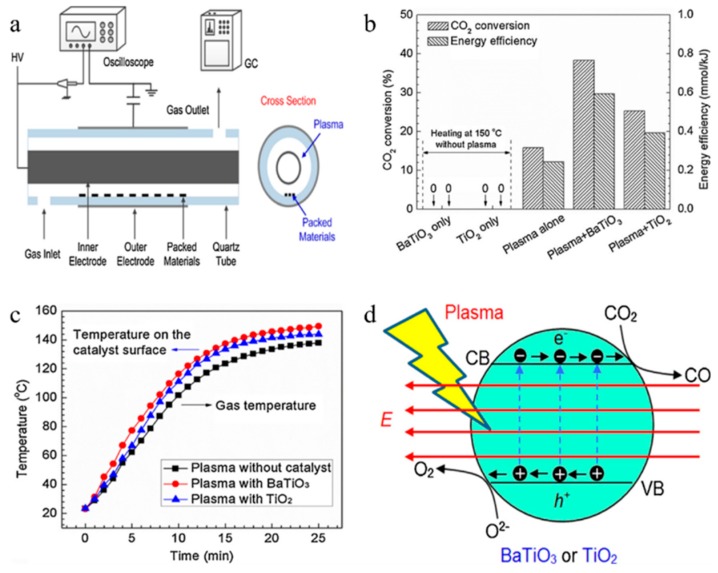
(**a**) diagram of reaction experimental device; (**b**) impact of plasma and catalyst on CO_2_ conversion; (**c**) changes in plasma gas temperature and catalyst surface temperature; (**d**) reaction mechanisms of photocatalytic decomposition of CO_2_ by plasma. Reproduced with permission from [101]; Copyright Elsevier, 2016.

**Figure 8 nanomaterials-09-01428-f008:**
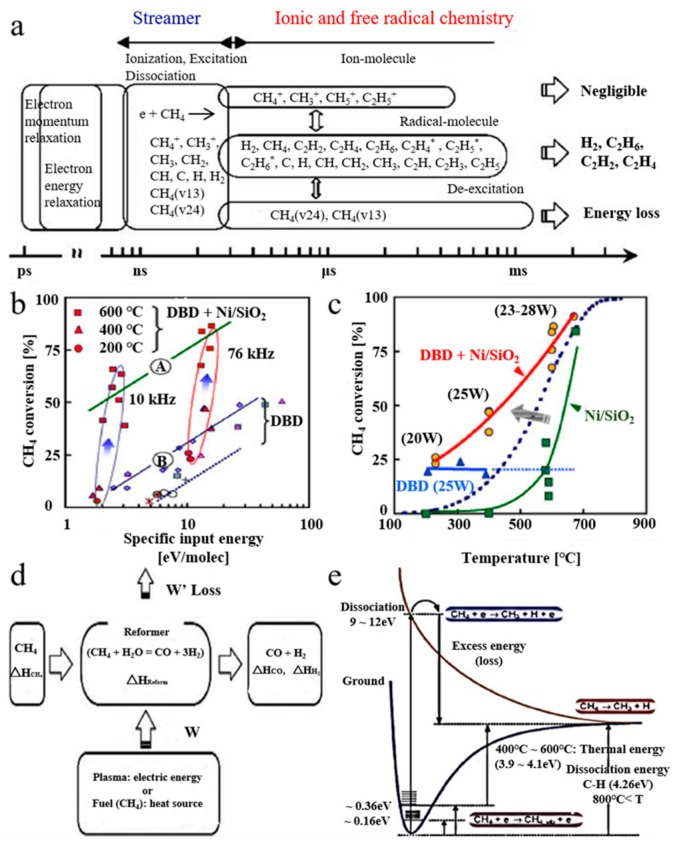
(**a**) reaction scheme; (**b**) CH_4_ conversion: (A) barrier discharge + Ni catalyst; (B) barrier discharge; (**c**) CH_4_ conversion; the blue dashed line represents the equilibrium conversion; (**d**) energy diagram; (**e**) methane activation. Reproduced with permission from [106]; Copyright Elsevier, 2004.

**Figure 9 nanomaterials-09-01428-f009:**
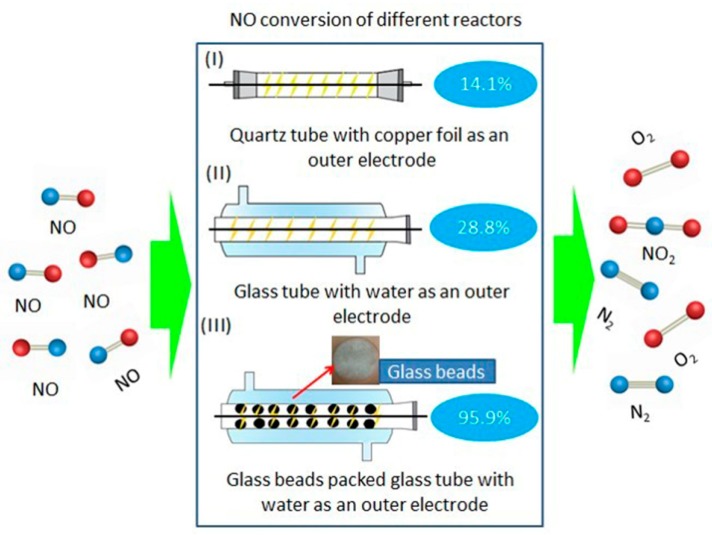
NO conversion under different reactors [109]; (Open Access).

**Figure 10 nanomaterials-09-01428-f010:**
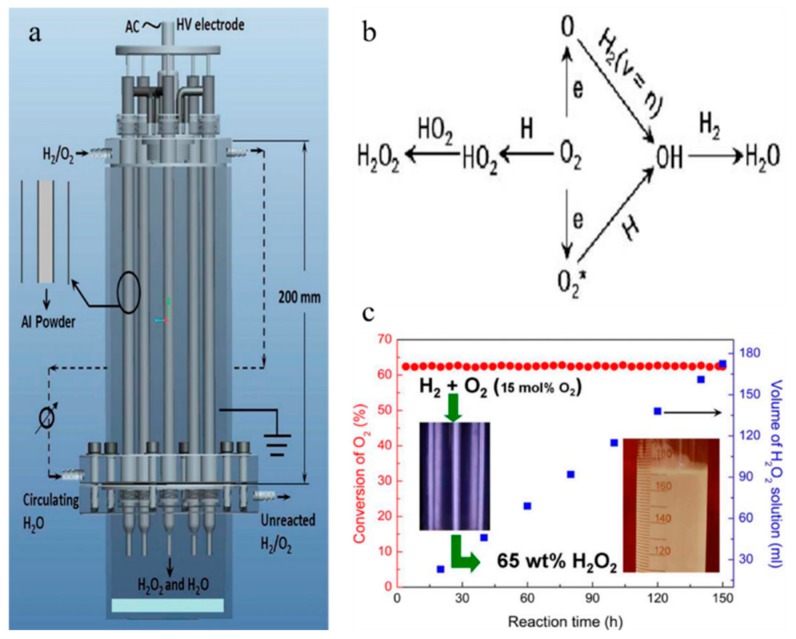
(**a**) diagram of double DBD reactor H_2_O_2_ synthesis device; (**b**) the main reactions network for formation of H_2_O_2_ and H_2_O in H_2_/O_2_ nonequilibrium plasma; (**c**) the relationship between O_2_ conversion, H_2_O_2_ product solution volume and reaction time. Reproduced with permission from [117]; Copyright Wiley, 2014.

**Figure 11 nanomaterials-09-01428-f011:**
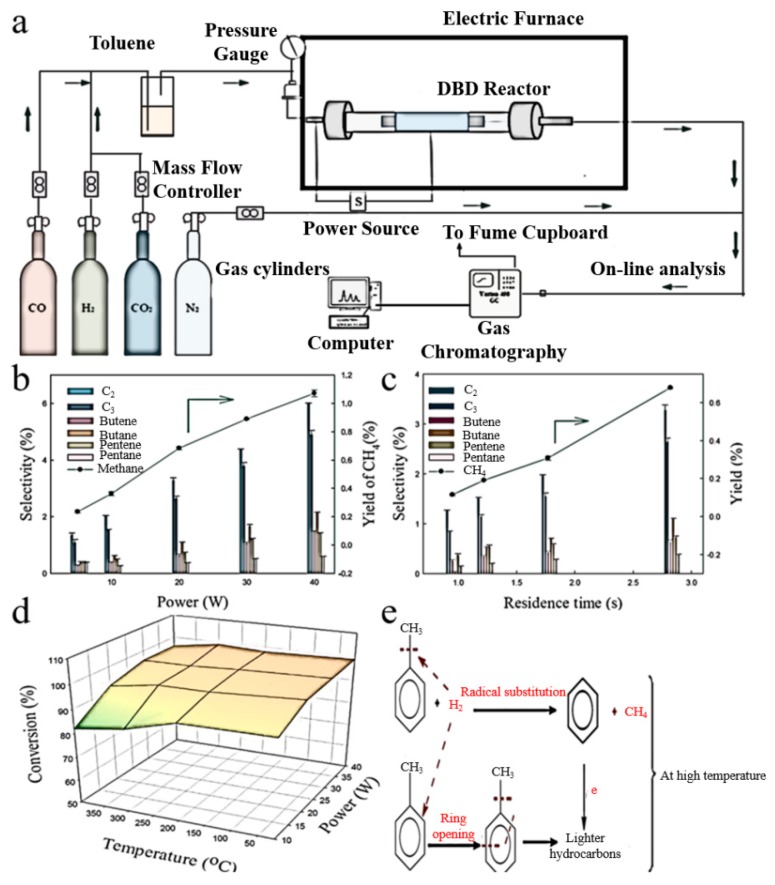
(**a**) experimental setup; (**b**) influence of plasma power on individual lower hydrocarbons; (**c**) effect of residence time on individual lower hydrocarbons; (**d**) influence of temperature on the toluene conversion; (**e**) reaction mechanism at elevated temperature. Reproduced with permission from [130]; Copyright Elsevier, 2019.

**Figure 12 nanomaterials-09-01428-f012:**
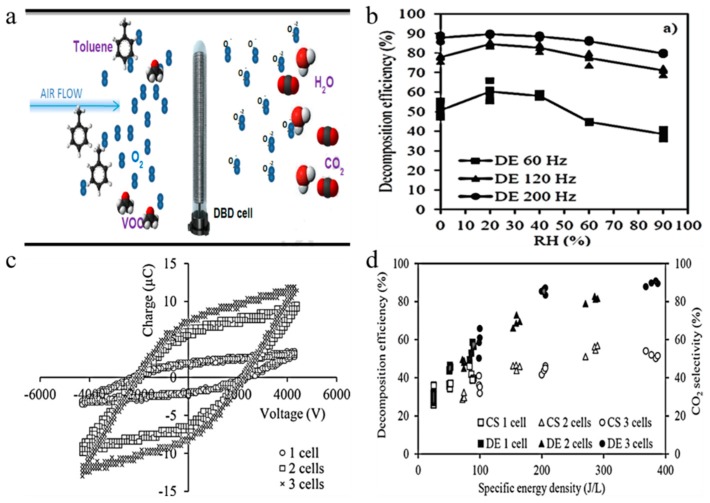
(**a**) graphical abstract; (**b**) the impact of relative humidity on decomposition efficiency at different frequencies; (**c**) Lissajous figures of different discharge regions; (**d**) effect of specific energy density on CO_2_ selectivity and decomposition efficiency. Reproduced with permission from [137]; Copyright Elsevier, 2018.

**Figure 13 nanomaterials-09-01428-f013:**
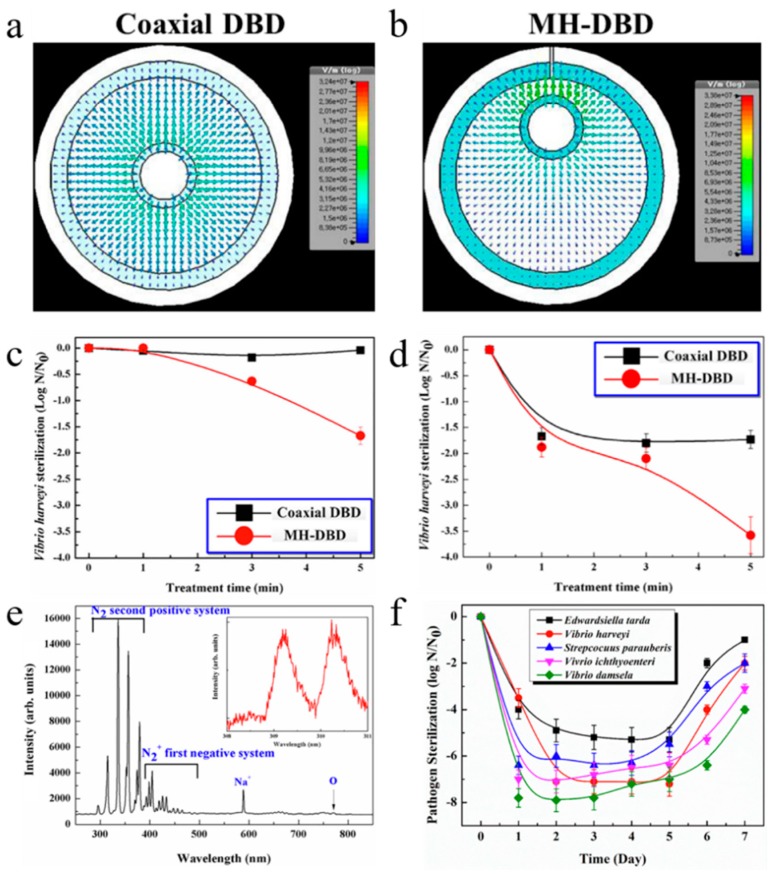
(**a**) electric filed distribution of DBD plasma; (**b**) electric filed distribution of MH-DBD plasma; (**c**) DBD and MH-DBD plasma sterilization curves comparison with nitrogen as the plasma forming gas; (**d**) DBD and MH-DBD plasma sterilization curves comparison with air as the plasma forming gas; (**e**) optical emission lines of air MH-DBD plasma; (**f**) pathogen sterilization curves. Reproduced with permission from [144]; Copyright Elsevier, 2019.

**Figure 14 nanomaterials-09-01428-f014:**
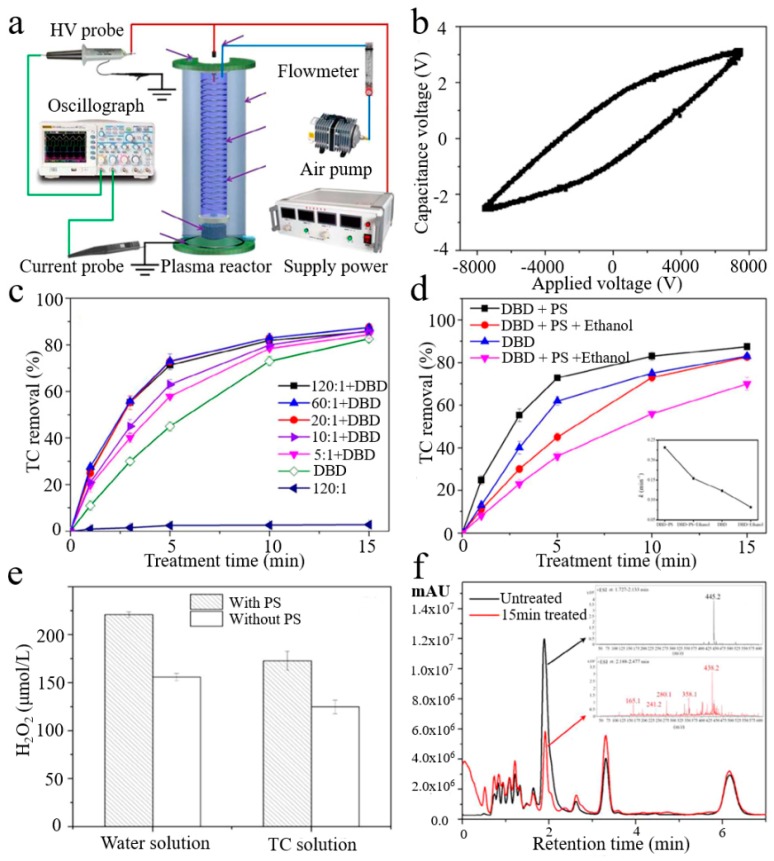
(**a**) gas phase surface discharge device diagram; (**b**) Lissajous figure; (**c**) effect of PS addition ratio on the TC removal; (**d**) effect of ethanol on TC removal; (**e**) effect of ethanol on H_2_O_2_ production; (**f**) total ion chromatograph and HPLC/MS chromatograph of TC decomposition intermediates. Reproduced with permission from [153]. Copyright Elsevier, 2018.

**Figure 15 nanomaterials-09-01428-f015:**
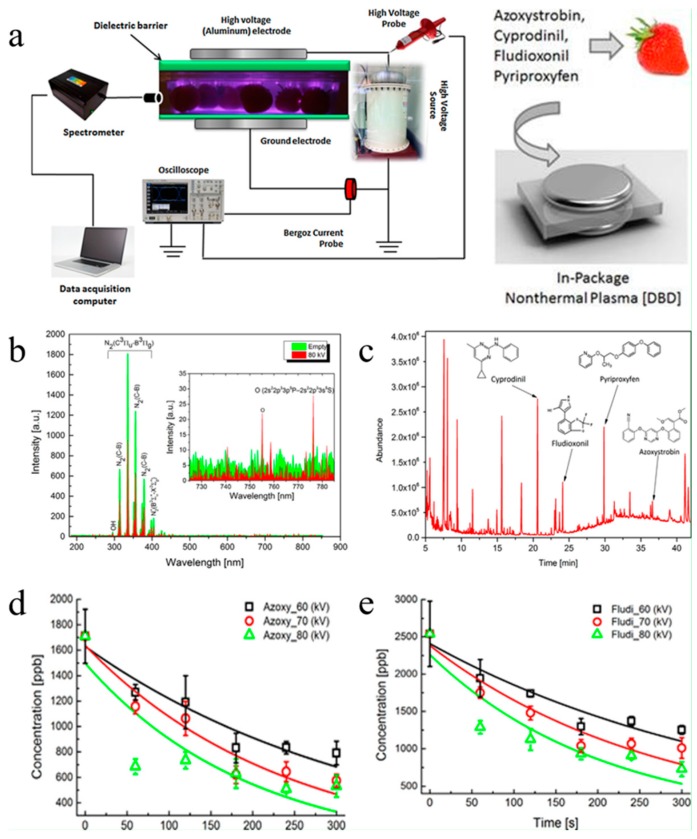
(**a**) the diagram of the DBD experimental device; (**b**) emission spectrum with or without strawberries; (**c**) retention times chromatogram of the four pesticides; (**d**) azoxystrobin concentrations before and after plasma treatment; (**e**) fludioxonil concentrations after plasma treatment. Reproduced with permission from [162]; Copyright Elsevier, 2014.

**Table 1 nanomaterials-09-01428-t001:** Highly dispersed catalysts prepared by a plasma technique.

Types of Catalyst	Types of Plasma	Characteristics	Remarks	Applications	References
Pt and Co catalyst	DBD plasma H_2_/N_2_/CH_4_	Higher selectivity		Methane conversion	[16]
Pt/TiO_2_ catalyst	DBD cold plasma Ar & H_2_	Smaller Pt particles homogeneously distributed	Size of particles: 1.7 nm	Mesoporous photocatalyst with enhanced activity	[17]
Pd/Graphene sheets	DBD plasma-assisted H_2_	Well-dispersed and higher catalytic efficiency	Size of particles: 2 nm	The hydrodesulfu-rization of carbonyl sulfide	[25]
Au/TiO_2_	DBD plasma O_2_	Higher activity small particle size and narrow size distribution	Size of particles: 2.6 nm	CO oxidation	[26]
Au/P25-PC	AP DBD cold plasma Ar & H_2_	Smaller size and higher CO oxidation activity	Size of particles: 4.6 nm	CO oxidation	[27]
Ni/MgO	DBD plasma H_2_	Smaller particle size and higher specific surface area	Size of particles: 3.4 nm	CO_2_ reforming of methane	[28]
Ni/Ce/SBA-15-P	DBD plasma Ar & H_2_	High specific surface area and good high-temperature stability	Size of particles: 7.1 nm	Methanation of carbon oxides	[29]
Ni/SiO_2_	Ammonia impregnation combined with DBD NH_3_	High dispersion and high temperature stability	Size of particles: 11.2 nm	CO methanation	[30]
Ni/ZrO_2_	DBD plasma H_2_	Highly dispersed and long-time stability	Size of particles: 10.6 nm	CO methanation	[31]
Ni/MgO	DBD plasma air	Low temperature activity and good stability	Size of particles: 5.4 nm	CO_2_ reforming of methane	[32]
Ni/MgAl_2_O_4_	DBD plasma air	High dispersion and unique structure	Size of particles: 8.9 nm	CO_2_ methanation	[33]
Ni/PVMT	DBD plasma H_2_	Ultralow Ni loading and excellent catalytic performance		CO methanation	[34]
Ni/SiO_2_	DBD plasma air	Low activity with small size and coke resistance	Size of particles: 7.1 nm	Steam reforming of methane	[35]
MnO	Wire-plate DBD O_2_	Low cost, environmentally friendly and high activity		Toluene decomposition	[36]
Co/CNTs	DBD plasma Ar	Uniform and dispersed Co particles		FTs	[37]
W_2_N	DBD plasma NH_3_	Excellent chemical stability and high performance	Size of particles: 5.99 nm	Acetylene hydrochlorina-tion	[38]
TiO_2_-β zeolite	DBD plasma O_2_	Higher methanol photodegradation rates	The BET specific surface area: 421.9 m^2^/g	Methanol photooxidatin	[39]

**Table 2 nanomaterials-09-01428-t002:** Defect-rich catalysts prepared by a plasma technique.

Types of Catalyst	Types of Plasma	Characteristics	Remarks	Applications	References
Multi-wall CNTs	DBD plasma O_2_	Generate oxygen-containing group	The BET specific surface area: 156 m^2^/g	Surface modification	[50]
Graphite	DBD plasma air	Create novel defects	Protrusions 1−5 nm in diameter and smooth depressions 5−7 nm wide	Modification of graphite surfaces	[51]
Co_3_O_4_ nanosheets	Plasma-engraved Ar	With high surface area and oxygen vacancies	The BET specific surface area: 160.26 m^2^/g	OER	[52]
Ag catalysts	Plasma-activated H_2_, Ar & O_2_	Lower overpotential	Pore-like defects 50−100 nm in size	Carbon dioxide electroreduction	[53]
Ar-TiO_2_	DBD plasma Ar	Oxygen vacancies and Ti^3+^ defects	The energy band gap reduction	Photocatalytic degradation of organic dyes	[54]
Teflon films	DBD H_2_O	Point electrode		Prevent degradation and erosion	[55]

**Table 3 nanomaterials-09-01428-t003:** Heteroatom-doped catalysts prepared by a plasma technique.

Types of Catalyst	Types of Plasma	Characteristics	Remarks	Applications	References
N-doped ZnO thin films	DBD pulsed laser deposition N_2_	Low temperature photoluminescence spectra	The maximum hole density: 10^−17^–10^−18^·cm^−3^	Make ZnO based electronic devices	[56]
N-PEGO	DBD plasma CO_2_ & NH_3_	High onset potential and good electrocatalysis stability	The BET specific surface area: 380.0 m^2^/g	ORR	[57]
N-vacancy-doped g-C_3_N_4_	DBD plasma H_2_	Outstanding photocatalytic		H_2_O_2_ production	[58]
Cu_2_O/N-TiO_2_	DBD plasma N_2_/Ar	Dispersed uniformly and higher photodegradation efficiency		Improved photocatalytic activity	[59]
S-doped g-C_3_N_4_	DBD plasma H_2_S	Larger specific surface area	The BET specific surface area: 52.8 m^2^/g	Enhanced photocatalytic performance	[60]
P-Co_3_O_4_	Ar plasma	Porous, discontinuous, and loose surface		HER and OER	[61]

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
