# Peer review of "A Review of Recent Advances of Dielectric Barrier Discharge Plasma in Catalysis"

_nanomaterials, 2019, doi:10.3390/nano9101428_

Round 1

Reviewer 1 Report

The manuscript reviews current advances of the application of dielectric barrier discharge plasmas in catalysis. The manuscript is quite detailed and comprehensive. The large number of citations highlights the importance of using DBD plasma as a cost effective technology in catalysis. Therefore, I would expect this manuscript to be of interest for a wide spectrum of readers. The manuscript is well structured and written with sufficient clarity. I have no objection for this manuscript to be published in Nanomaterials after the authors address the following comments:

A few figures have poor resolution and therefore lack clarity. I suggest those figure should be replaced with ones having better resolution, e.g. Figure 3, 6, 8, 11, 12, and 14. In table 1, 2 and 3, for the columns under the name “Types of plasma” the name of the processing gas should be mentioned clearly for each study. Figure 1 caption does not fully cover the content of the figure, please update. Typos or grammatical errors present in the following lines, please correct: 53, 160, 188, 284, 288, 541, 576, 594, 644, 664, 667, 723, 764 For sections: 2.3.1, 2.3.2, 2.3.3 please write the full name of the atom rather than the symbol. The abbreviations in the following lines: 199, 220, 277 should be defined. The names of bacteria escherichia coli and bacillus subtilis in lines 647 and 650, should be italic. Journal names in the reference list should be abbreviated.

Author Response

The figures that have poor resolution in the text have been replaced or become clear through processing,  e.g. Figure 3, 6, 8, 11, 12, and 14. In table 1, 2 and 3, for the columns under the name “Types of plasma” the name of the processing gas has been clearly mentioned and is added to the table. Figure 1 caption has been modified and completely covers the contents of the figure. e.g. Figure 1. Two typical DBD electrode devices: (a) planar reactor; (b) cylindrical reactor is changed to Figure 1. Two typical DBD electrode devices: (a) planar reactor; (b) cylindrical reactor; (c) two discharge types of surface discharge and coplanar discharge. Typos or grammatical errors present in the following lines,which has been corrected:53 use is changed to usedï¼›160 aiti-sintering is changed to anti-sinteringï¼›188 Thence is changed to Henceï¼›284 Plasma catalysis synergy is changed to Plasma synergistic catalysisï¼›541 grounded is changed to groundï¼›576 product is changed to productsï¼›667 helps identify is changed to helps to identifyï¼›764 micropollutant is changed to micropollutants and 1 is changed to oneï¼›For sections: 2.3.1, 2.3.2, 2.3.3,the full name of the atom has been written,e.g. N is changed to Nitrogenï¼›S  is changed to Sulphurï¼›P is changed to Phosphorus. The abbreviations in the following lines: 199, 220, 277 have been defined. e.g. ORR is oxygen reduction reactionï¼›OER is oxygen evolution reactionï¼›HER-OER is hydrogen evolution reaction and oxygen evolution reaction. The names of bacteria escherichia coli and bacillus subtilis in lines 647 and 650, have been changed to italics. Journal names in the reference list have been abbreviated.

Reviewer 2 Report

The paper describes comprehensive summary on dielectric barrier discharge plasma method in catalytic fields. It is a well written and informative paper with an adequate level. The subject matter is suitable for the Journal. The quality of the presentation is appropriate. It can be accepted after minor revisions.

Specific comments:

1. There are several typos which should be corrected.

2. English level needs to be improved.

Author Response

Typos or grammatical errors present in the following lines,which has been corrected:53 use is changed to usedï¼›160 aiti-sintering is changed to anti-sinteringï¼›188 Thence is changed to Henceï¼›284 Plasma catalysis synergy is changed to Plasma synergistic catalysisï¼›541 grounded is changed to groundï¼›576 product is changed to productsï¼›667 helps identify is changed to helps to identifyï¼›764 micropollutant is changed to micropollutants and 1 is changed to one.

Round 2

Reviewer 1 Report

The authors have performed satisfactory work towards improving the quality of the manuscript based on the suggested comments. Therefore, I can now recommended for publication in Nanomaterials.